# Meta-analysis of arterial anastomosis techniques in head and neck free tissue transfer

**Yu-Jing Wang[1], Xiu-Ling Wang[2], Shan Jin[3], Ran Zhang[3], Yu-Qin Gao[1]***

**1** Department of Nursing, School and Hospital of Stomatology, China Medical University, Shenyang, Liaoning Province, China, **2** Department of Cardiology, The First Hospital of China Medical University, Shenyang, Liaoning Province, China, **3** Department of Oral and Maxillofacial Surgery, School and Hospital of Stomatology, China Medical University, Shenyang, Liaoning Province, China

* cmuyuqingao@hotmail.com

**Data Availability Statement:** All relevant data are within the manuscript and its Supporting information files.

**Funding:** The authors received no specific funding for this work.

## Abstract

The present meta-analysis aimed to investigate the differences in the incidence of thrombosis and vascular compromise in arterial anastomosis between microvascular anastomotic devices and hand-sewn techniques during free tissue transfer in the head and neck. We searched for articles in PubMed/Medline, CNKI, WANFANG DATA, Cochrane Central Register of Controlled Trials, Cochrane Database of Systematic Reviews, and Web of Science, from January 1, 1962 till April 1, 2020 that reported data of microvascular anastomosis during free tissue transfer in the head and neck. The incidence of arterial thrombosis or vascular compromise, or both was the primary outcome. The secondary outcome was anastomotic time. We also assessed the sensitivity and the risk of bias. This meta-analysis included 583 arterial anastomoses from six studies. The group using microvascular anastomotic devices tended to have an increased incidence of arterial thrombosis and vascular compromise (risk ratio (RR), 3.42; $P$ = 0.38; 95% confidence interval (CI), 0.91–12.77). The hand-sewn technique took significantly longer to perform the anastomosis compared with that of the microvascular anastomotic devices (weighted mean difference, 15.26 min; $P$<0.01; 95% CI, 14.65–15.87). Microvascular anastomotic devices might increase the risk of arterial thrombosis and vascular compromise compared with the hand-sewn technique; however, further randomized controlled trials are needed to provide a more accurate estimate. The application of microvascular anastomotic devices will help to reduce anastomotic surgery time and achieve acceptable vessel opening, benefiting from the developments of arterial couplers and microsurgical techniques.

## Introduction

Microvascular free tissue transfers have achieved widespread acceptance as the gold standards to repair of complex defects in the head and neck [1, 2]. In the transfer of free tissue, the survival rate mainly depends on forming a patent anastomoses between the recipient and donor

**Competing interests:** The authors have declared that no competing interests exist.

vessels. Moreover, in the transfer of free tissue, one of the most technically challenging and crucial elements is microvascular anastomosis. Vascular anastomosis traditionally relies on hand-sewn techniques involving nylon microvascular sutures (8–0 to 10–0). In 1962, Nakayama et al. first described venous couplers comprising two metal rings and twelve interlocking pins with matching holes that could achieve a patent venous anastomosis [3]. As an alternative method to hand-sewn techniques, a microvascular anastomotic device became commercially available in the 1980s, known as the Unilink coupler [4]. As the thicker wall of artery is more difficult to be everted over the pins of the coupler, the microvascular anastomotic device is more suited to venous anastomosis rather than arterial anastomosis. Microvascular anastomotic devices have proved to be reliable and time-saving in venous anastomosis [5, 6]. In recent years, there have also been some reports about the clinical use of microvascular anastomotic devices in arterial anastomoses. However, to date, there have been no evidence-based reports that compared arterial anastomosis using couplers with hand-sewn sutures. Therefore, the present meta-analysis aimed to assess the quality of microvascular anastomotic device in arterial anastomosis and the evaluate statistically the difference in the incidence of vascular compromise, thrombosis, or both, and the anastomotic time between coupler and hand-sewn techniques.

## Materials and methods

This was a retrospective meta-analysis; therefore, the China Medical University institutional review board granted a written exemption from ethical review. The clinical enquiries were executed in accordance with the principles of the Declaration of Helsinki. The analysis protocol adhered strictly to the PRISMA guidelines (S1 File).

### Search strategy

Searches were performed in databases including CNKI, WANFANG DATA, Cochrane Central Register of Controlled Trials, Cochrane Database of Systematic Reviews, Web of Science, and PubMed/Medline, for articles from January 1, 1962 until April 1, 2020 that reported data concerning microvascular anastomosis in head and neck free tissue transfer. The search strategy was carried out using broad key terms in all fields as follows: ("microvascular anastomotic device" OR "microvascular coupler" OR "coupler") AND ("head and neck" OR "head and neck reconstruction"). The publication languages were limited to English and Chinese. The search was conducted using both English and Chinese search terms. All relevant studies were included as a result of manual searches of the reference lists of the retrieved papers.

### Inclusion and exclusion criteria

The following inclusion criteria were used:

1. Studies of head and neck reconstruction involving free tissue transfer.

2. Studies that included data of the comparison between a microvascular anastomotic device and the hand-sewn technique for arterial anastomosis.

3. Studies in which at least one of the following variables was regarded as the primary outcome: The arterial thrombosis incidence, the incidence of vascular compromise, and anastomotic time.

4. Studies from different periods by the same department were regarded as separate studies and included.

The following exclusion criteria were adopted:

1. Studies or case series that only reported the application of a microvascular anastomotic device or the hand-sewn technique.

2. Single case reports, studies with a sample size less than 10, or animal model studies.

3. Reviews, comments, letters to the editor, or conference papers.

4. Reports not written in Chinese or English.

## Extracting the data, and assessing its quality and bias

Two authors (W.Y.J., J.S.) independently extracted the data and assessed the bias. Disagreements were resolved by a third author (Z.R.). A predefined electronic form was used to record, verify, and document the extracted data. For comparison, we collected the following data:

1. Baseline data of the study (author, journal and publication year).

2. Patient demographics (sample size, sex, age, and anatomical region of reconstruction).

3. The design of the study (type of study, the number of arterial anastomoses, and the anastomotic technique).

4. Intraoperative and postoperative outcome measures (vascular thrombosis, vascular compromise or arterial spasm, time taken to complete the arterial anastomosis, and complications associated with the coupler).

To assess the quality of the methods used in the included studies, we used the Newcastle-Ottawa Scale (NOS), in which moderate to high quality was indicated by five stars.

The ROBINS-I tool [7] was used to assess the risk of bias in non-randomized studies. Bias risk was categorized as low (L), moderate (M), serious (S), critical (C), or no information (NI). ROBINS-I assesses the risk of bias in seven bias domains: Bias caused by confounding data, biased selection of study participants, biased intervention classification, bias caused by digressions from the intended interventions, bias resulting from missing data, biased outcome measurement, and biased selection of which results to report.

## Statistical analysis

Data analysis and synthesis were performed using statistical software R (R-4.0.0). The estimated by risk ratio (RR) value and its associated 95% confidence interval (CI) were used to assess the strength of the association between arterial anastomosis and vascular thrombosis and vascular compromise. The weighted mean difference (WMD) value and its associated 95% CI were used to compare the anastomose time between the microvascular anastomotic devices and the hand-sewn technique. The significance of the pooled RR was determined using a Z-test, with statistical significance being set at $P<0.05$. A forest plot was used to display the results of the meta-analysis [8]. Cochran's Q and $I^2$ statistic [9, 10] were used to assess the heterogeneity of all included studies. If the $P$ value was $> 0.10$ or the $I^2$ value of was $< 50\%$, we used a fixed effects model, calculated using the Mantel-Haenszel (M-H) method. Otherwise, as a result of the anticipated study heterogeneity, a random effects model was used. The Begg rank correlation test was used to assess the extent of publication bias [8, 11], and $P<0.05$ was considered as significant publication bias. In addition, sensitivity analysis was performed using STATA 11.0 (StataCorp, College Station, TX, USA).

## Results

The search strategy identified 158 studies, which were imported into an EndNote X8-based bibliographic database. Forty-two studies were excluded after removal of duplicates. Screening the titles and abstracts excluded a further 68 studies because they were: Not completely relevant (n = 54), review articles (n = 5), case reports (n = 4), letters to the editor (n = 2), conference papers (n = 1), written in Italian (n = 1), or written in French (n = 1). Then, we read the full text of the remaining 48 articles and assessed them for eligibility, which resulted in exclusion for the following reasons: No comparison between a microvascular anastomotic device and the hand-sewn technique (n = 25), only venous anastomoses data (n = 9), unable to extract comparison data (n = 5), no clear outcome measures (n = 1), and a sample size less than 10 (n = 2). Finally, six studies that contained sufficient and direct comparison data were included in this meta-analysis. The PRISMA guidelines were followed, and the study screening was illustrated using the PRISMA flow diagram (Fig 1).

### Characteristics of included studies

The six included studies [12–17] were all retrospective cohorts. There were four English language articles and two Chinese language articles. A total of 583 arterial end-to-end anastomoses were performed, including 251 arteries anastomosed using microvascular anastomotic devices and 332 arteries anastomosed using hand-sewn techniques. Table 1 shows the characteristics of the studies that we included. Experienced surgeons who had undergone strict microsurgery training performed all the microvascular anastomoses. Standard clinical examinations (pin-prick testing, palpation, and flap color) were used to monitor the free flaps. In cases where vascular compromise was suspected, emergency exploration would be performed. The included studies were of moderate or high quality, with at least five stars from the NOS analysis (S2 File).

### Main results of the meta-analysis

**Incidence of arterial thrombosis and vascular compromise.** All six studies reported arterial thrombosis and vascular compromise rates between 0% and 11.8%. A direct comparison meta-analysis was performed using the pooled risk estimates from the six studies, which identified a higher incidence of arterial thrombosis and vascular compromise for the microvascular anastomotic device group (RR, 3.42). However, the difference was not statistically significant ($P = 0.38$; 95% CI, 0.91–12.77; Fig 2). This result indicated that using a microvascular anastomotic device for arterial anastomosis might cause a higher incidence of vascular compromise or thrombosis compared with that of the hand-sewn technique.

**Anastomotic time of microvascular anastomotic devices versus the hand-sewn technique.** Most of the studies provided approximate statistical data concerning the anastomotic time. Only two studies [16, 17] performed exact statistics on the anastomotic time via a direct comparison of the time taken for anastomosis using the microvascular anastomotic device with that of the hand-sewn technique. The result indicated that in the two studies, the hand-sewn technique took a significantly longer time compared with that of the microvascular anastomotic device (WMD, 15.26 min; $P < 0.01$; 95% CI, 14.65–15.87, Fig 3).

### Risk of bias and sensitivity analysis

According to the ROBIN-S assessment, there was a serious risk of overall bias in the six studies included in this meta-analysis (Table 2). The essential limitations of retrospective studies

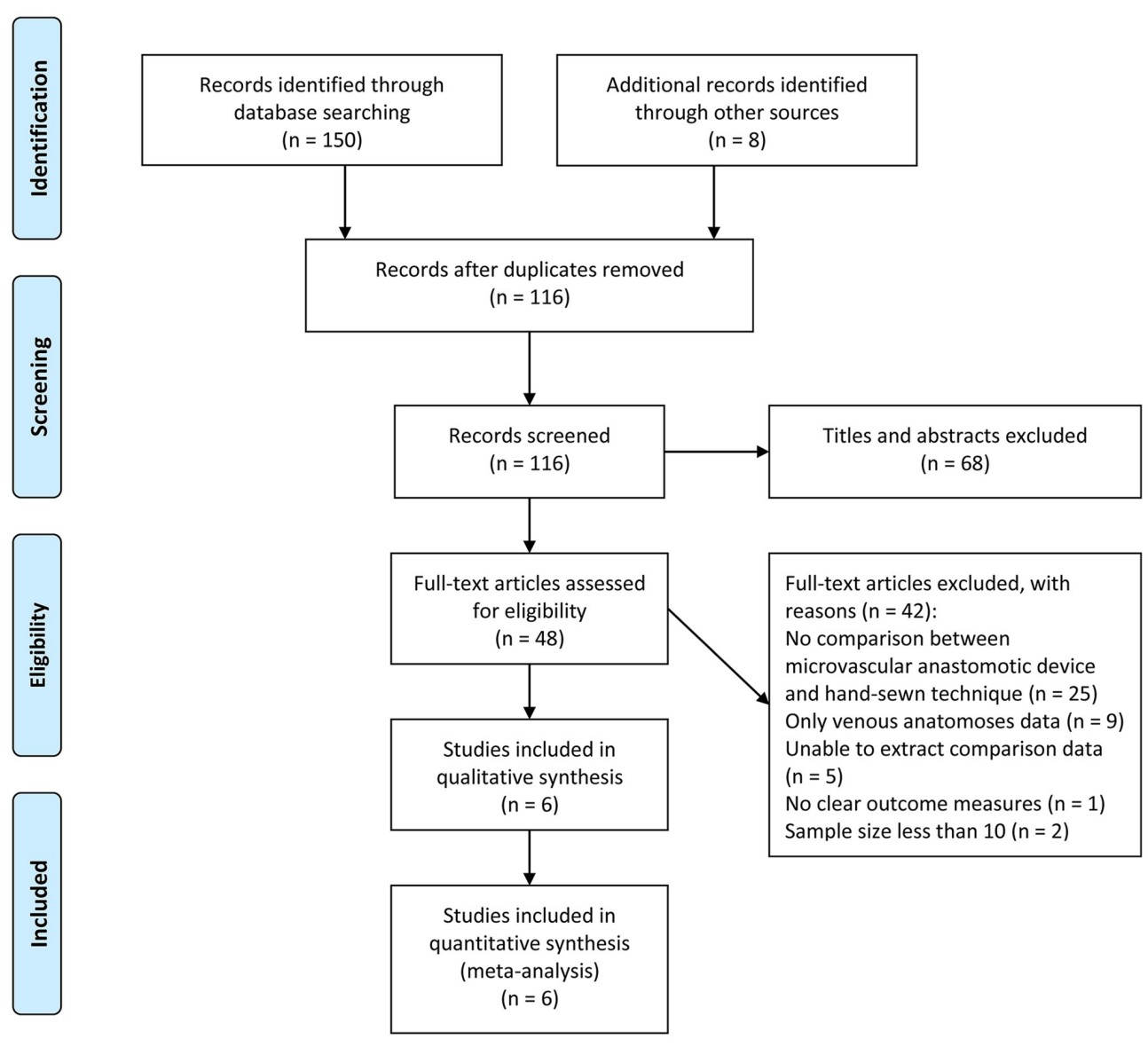

**Fig 1. PRISMA flow diagram for the study selection process.**

**Table 1. Study characteristics in this meta-analysis.**

| First author | Country | Year of publication | Study design | Study period | No. of arterial coupler anastomoses | No. of arterial hand-sewn anastomoses |
|---|---|---|---|---|---|---|
| Maisie L. Shindo | USA | 1996 | Retrospective | 1992–1995 | 17 | 62 |
| Natalya Chernichenko | USA | 2008 | Retrospective | 2001–2006 | 124 | 3 |
| Wang | China | 2015 | Retrospective | 2013–2014 | 7 | 57 |
| Sun | China | 2016 | Retrospective | 2014–2015 | 3 | 18 |
| Z.Y. Yang | China | 2016 | Retrospective | 2015–2016 | 44 | 125 |
| Guo | China | 2019 | Retrospective | 2016–2018 | 56 | 67 |

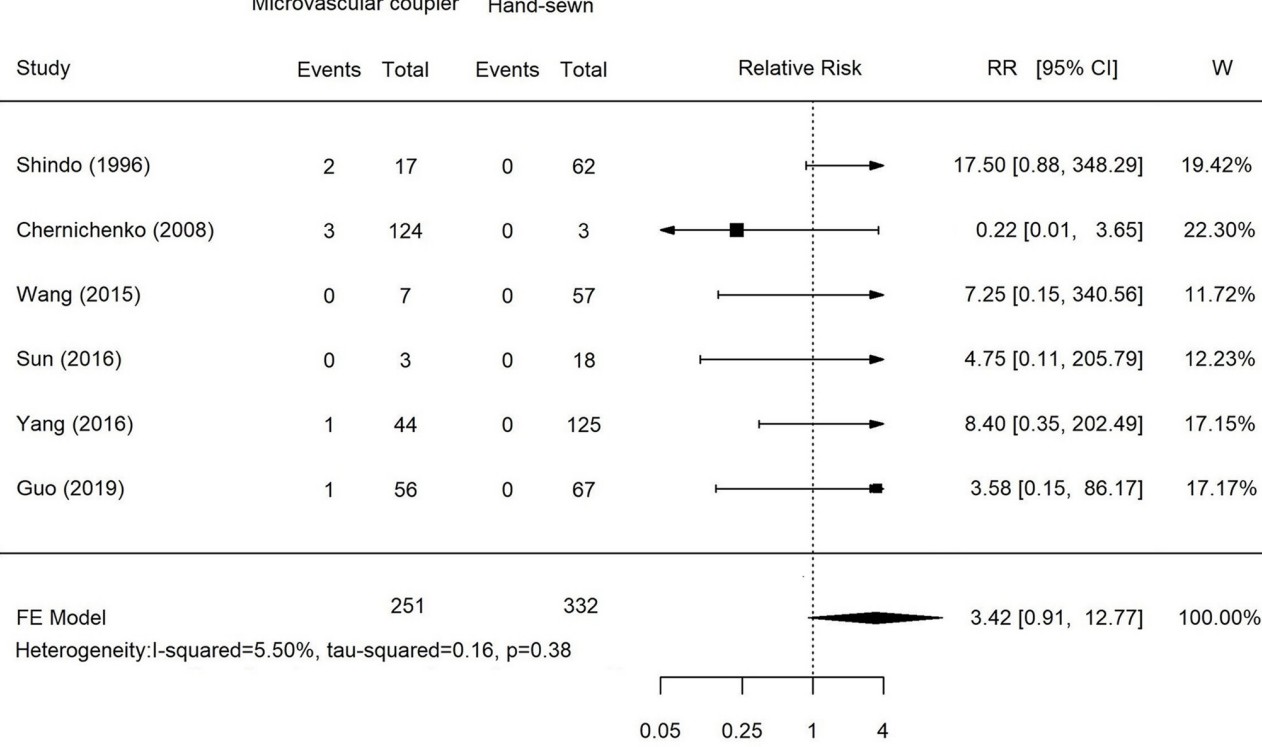

**Fig 2. Meta-analysis of the incidence of arterial thrombosis and vascular compromise between the microvascular coupler and hand-sewn techniques.**

meant that the majority of included studies were at serious risk of bias because of confounding data, selection of participants, and deviations from intended interventions.

According to the Begg rank correlation test, this meta-analysis contained no obvious publication bias ($P$ = 0.72, Fig 4).

A sensitivity analysis was performed to detect the influence on the pooled result of removing single studies one at a time, which showed that no single study interfered with the overall results of this meta-analysis (Fig 5).

## Discussion

Reconstruction involving head and neck free tissue transfers have become increasing popular in recent decades, largely because of the reliability and quality of the anastomosis. Compared with hand-sewn techniques, microvascular anastomotic devices have gained popularity

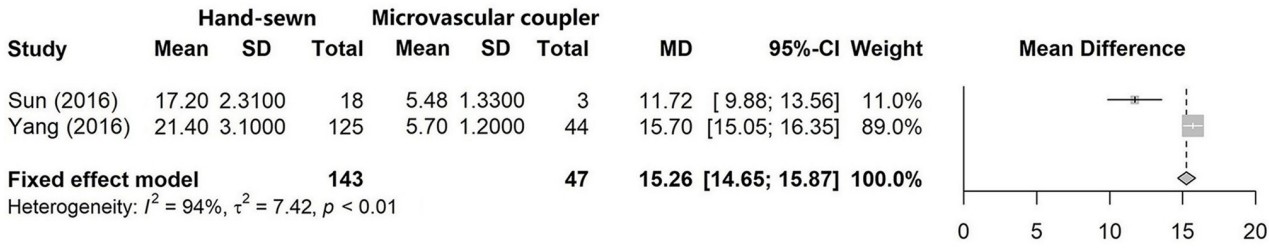

**Fig 3. Publication bias as revealed using a Begg's funnel plot.**

**Table 2. ROBINS-I quality assessment of the characteristics of the included studies.**

| Study | Risk of bias* | | | | | | | |
|---|---|---|---|---|---|---|---|---|
| | **A** | **B** | **C** | **D** | **E** | **F** | **G** | **Overall** |
| Shindo 1996 | S | S | S | S | M | S | M | S |
| Chernichenko 2008 | S | S | S | S | S | S | M | S |
| Wang 2015 | S | S | M | S | M | S | S | S |
| Sun 2016 | S | S | S | S | M | S | S | S |
| Yang 2016 | S | S | S | S | M | M | M | S |
| Guo 2019 | S | S | S | S | M | S | M | S |

*Risk of bias was assessed using the ROBINS-I tool and classified as low (L), moderate (M), serious (S), critical (C), or no information (NI) for each domain of bias: Confounding (A), Selection of participants (B), Classification of interventions (C), Deviations from intended interventions (D), Missing data (E), Measurement of outcomes (F), Selection of reported results (G), Overall bias.

because of their simplicity, reliability, and speed, especially in venous anastomosis [18, 19]. Wain et al. [20] found that compared with a sutured arterial anastomosis, a coupled one was less likely to induce thrombogenesis, as evidence by reduced wall shear stress revealed using a computational fluid dynamic model.

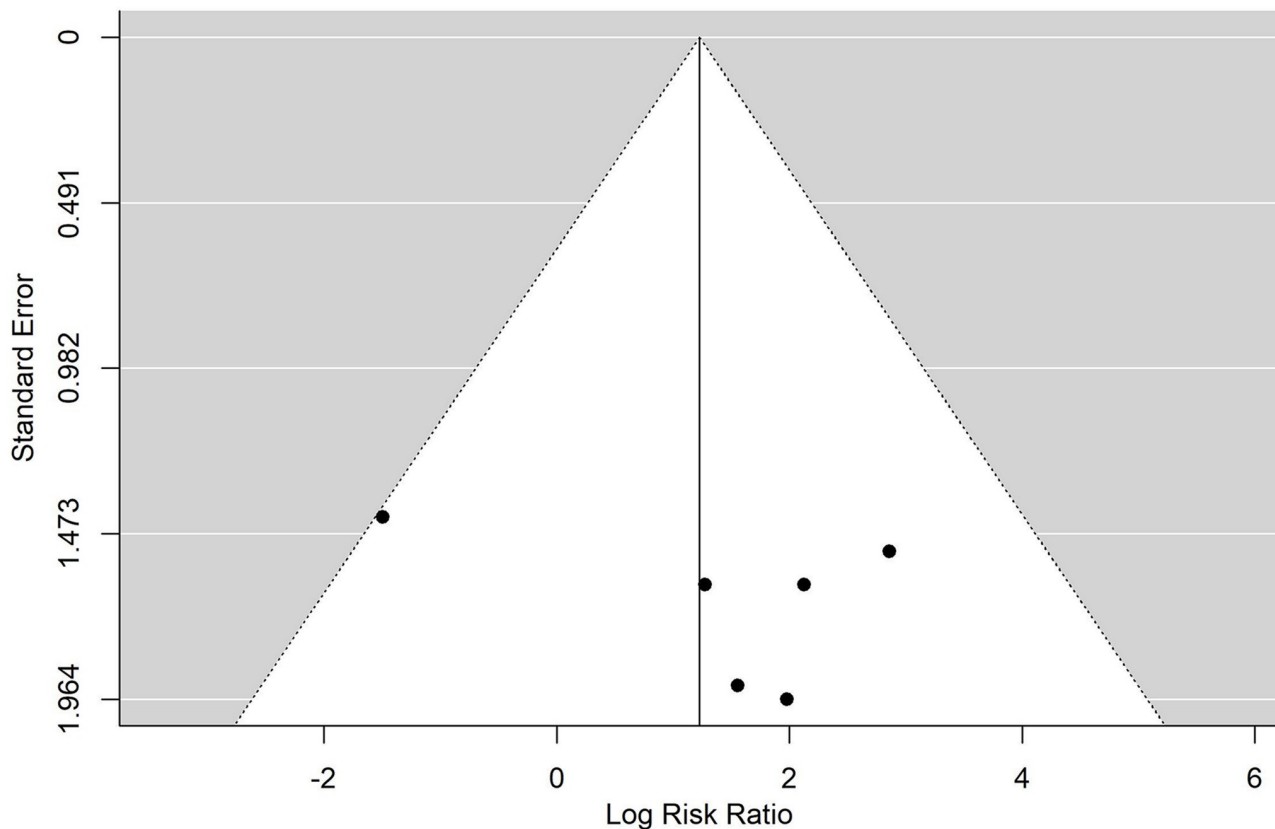

**Fig 4. Meta-analysis of microvascular coupler versus hand-sewn techniques for anastomotic time.**

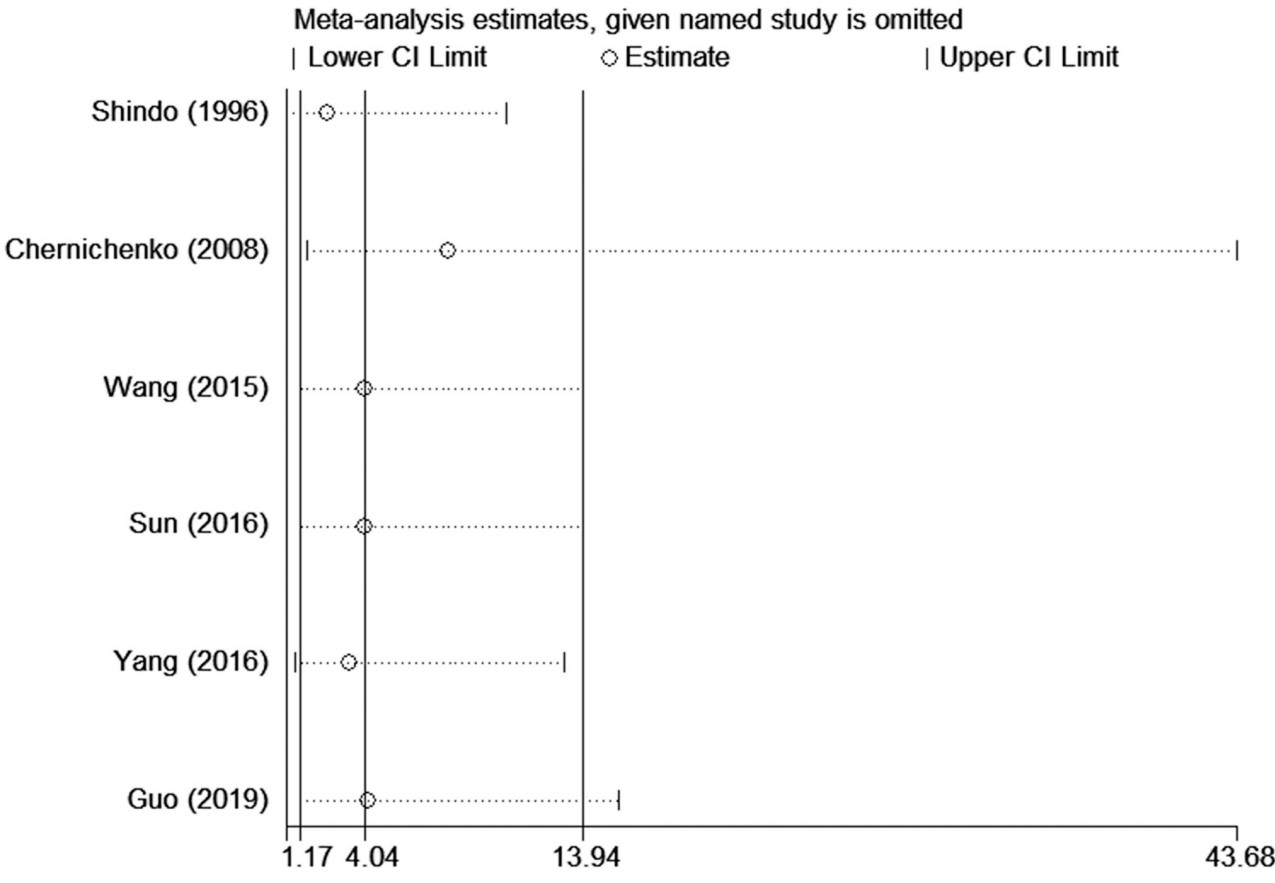

**Fig 5. Results of the sensitivity analysis.**

Although the effectiveness and safety of microvascular anastomotic devices in venous anastomosis were widely reported, using mechanical coupling devices for arterial anastomoses remains controversial [21]. This likely originated from the high incidence of arterial thrombosis reported by a few case series in the early 1990s [12, 22]. These findings were consistent with reports of microvascular anastomotic device use in the reconstruction of other sites [23, 24]. Consequently, most head and neck reconstructive surgeons do not routinely apply microvascular couplers for arterial anastomoses for the following reasons: First, the vessel walls of arteries in the head and neck region are much thicker than those of the veins, and lack elasticity to allow their dilation during vessel preparation. Many authors have noted difficulties manipulating the thick artery walls. Inadequate vessel eversion might have led to obstruction of the arterial lumen and decreased the laminar blood flow. Second, the recipient vessels in the head and neck region are vulnerable to the effects of preoperative radiotherapy. The radiated vessels often exhibit fibrosis and are easy to tear. Some authors suggested that in patients who had received radiation therapy, arterial coupling should not be attempted [25]. Third, elderly patients or patients with hypertension might be more prone to atherosclerosis, which also increases the technical difficulties or clinically relevant complications. Last, in contrast to veins, the arterial wall is less elastic. If a discrepancy between the calibers of the recipient and donor arteries is encountered, the use of a microvascular anastomotic device would probably be abandoned.

In practice, preoperative radiation therapy is not an absolute contraindication for use of an arterial coupler. If there is an obvious atherosclerotic plaque or fibrosis within the artery, surgeons could try to resect the artery to reveal a section with minimal hardening [26]. Thus, it is important to preserve an adequate length of the recipient artery in patients with atherosclerosis or who have received radiation. According to a review [25], the indications for the application of microvascular anastomotic devices in arterial anastomosis are as following: First, the caliber of the recipient and donor arteries should be no less than 1 mm. Second, the luminal diameters of the recipient and donor arteries should be discrepant by not more than a 1.5:1 ratio. Third, there is no severe fibrosis or atherosclerotic plaques within the artery.

Over the past two decades, microvascular anastomotic devices have gradually been improved and now offer unique advantages over the classical suture technique for arterial anastomosis. Ross et al. [27] and Chen et al. [28] championed arterial coupling, describing the incidence of thrombosis in only 1/50 (2%) and 1/45 (2%) cases, respectively. Some authors [26, 29] also reported the utilization of microvascular couplers in the salvage setting for free flap surgery in the head and neck region.

In terms of anastomotic time, microvascular anastomotic devices have absolute superiority over the hand-sewn technique. The time taken to compete an anastomosis was reduced by an average of 8–19 when using a microvascular anastomotic device compared with that using the hand-sewn technique [15, 19, 30]. However, a microvascular anastomotic device could cost up to 20 times as much as a suture set, which is another drawback of the coupler device. However, a cost-benefit analysis demonstrated that the overall cost of using a coupler device decreased, mainly because of the reduced overall operation time [31].

This meta-analysis had several limitations. The included studies were all retrospective in nature, placing them at risk of a variety of biases, particularly confounding bias. An imbalanced distribution of patients in the microvascular anastomotic device and hand-sewn technique groups from the included studies might also have resulted in bias of this meta-analysis. Arterial thrombosis was a low rate event in in the included studies; therefore, a prospective cohort study with a large sample size is required. In most of the included studies, the patient inclusion and exclusion criteria were not clear. Flap survival might be influenced by different individual risk factors and co-morbidities [32]. Moreover, most of the included studies had missing explicit and consistent data records, making it difficult to carry out a direct comparison between the microvascular anastomotic group and the hand-sewn group. Thus, further randomized controlled trials are needed to provide a more accurate estimate.

In conclusion, the results from the present study showed that microvascular anastomotic devices might increase the risk of arterial thrombosis and vascular compromise compared with that of the hand-sewn technique. However, the differences were not statistically significant. The application of microvascular anastomotic devices will help to decrease the anastomotic time. Further developments of arterial couplers and microsurgical techniques are needed to achieve more satisfactory vessel patency.

## Supporting information

**S1 File. PRISMA 2009 checklist.**
(DOC)

**S2 File. Assessment of the quality of the studies included in the meta-analysis.**
(DOCX)

## Author Contributions

**Conceptualization:** Yu-Jing Wang, Yu-Qin Gao.

**Data curation:** Yu-Qin Gao.

**Formal analysis:** Yu-Jing Wang, Shan Jin, Ran Zhang.

**Investigation:** Yu-Jing Wang, Xiu-Ling Wang, Shan Jin, Ran Zhang.

**Methodology:** Yu-Jing Wang, Xiu-Ling Wang, Shan Jin, Ran Zhang.

**Project administration:** Yu-Qin Gao.

**Resources:** Yu-Qin Gao.

**Supervision:** Yu-Qin Gao.

**Writing – original draft:** Yu-Jing Wang.

**Writing – review & editing:** Yu-Jing Wang.

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
