## [Decision Letter · Decision Letter 0]

1 Feb 2021

PONE-D-20-38890

Meta-analysis of arterial anastomosis techniques in head and neck free tissue transfer

PLOS ONE

Dear Dr. Gao,

Thank you for submitting your manuscript to PLOS ONE. After careful consideration, we feel that it has merit but does not fully meet PLOS ONE’s publication criteria as it currently stands. Therefore, we invite you to submit a revised version of the manuscript that addresses the points raised during the review process.

We look forward to receiving your revised manuscript.

Kind regards,

Peter Dziegielewski, MD, FRCSC

Academic Editor

PLOS ONE

Additional Editor Comments:

Academic Editor: I agree with the reviewers. Please make the changes requested by the reviewers.

Also please address the following questions:

- was there any data on which arteries were used as donor arteries in the neck? Are some better than others?

- was there any data t compare types of flaps used?

- was there any data on when the arterial thrombosis occurred? Immediate? < 24 hrs? > 24 hrs?

- was there any data on the size of coupler used in failed anastomoses? 

Journal Requirements:

2.) Thank you for submitting the above manuscript to PLOS ONE. During our internal evaluation of the manuscript, we found significant text overlap between your submission and the following previously published works, some of which you are an author.

https://www.jprasurg.com/article/S1748-6815(19)30521-2/fulltext

https://onlinelibrary.wiley.com/doi/abs/10.1002/hed.26139

https://www.bjoms.com/article/S0266-4356(19)30748-X/fulltext

https://journals.plos.org/plosone/article?id=10.1371%2Fjournal.pone.0134805

Please revise the manuscript to rephrase the duplicated text, cite your sources, and provide details as to how the current manuscript advances on previous work. Please note that further consideration is dependent on the submission of a manuscript that addresses these concerns about the overlap in text with published work.

Reviewers' comments:

Reviewer's Responses to Questions

**Comments to the Author**

1. Is the manuscript technically sound, and do the data support the conclusions?

Reviewer #1: Yes

Reviewer #2: Yes

2. Has the statistical analysis been performed appropriately and rigorously? 

Reviewer #1: Yes

Reviewer #2: Yes

3. Have the authors made all data underlying the findings in their manuscript fully available?

Reviewer #1: Yes

Reviewer #2: Yes

4. Is the manuscript presented in an intelligible fashion and written in standard English?

Reviewer #1: Yes

Reviewer #2: Yes

5. Review Comments to the Author

Reviewer #1: Wang et al. perform a meta-analysis comparing thrombosis outcomes in suture versus coupler devices in arterial anastomoses for free flaps for head and neck reconstruction. Overall, the authors have sound methodology for their meta-analysis and their question is appropriately focused for a meta-analysis. Their discussion and analysis is overall appropriate.

1. My primary concern is the imbalance of patient distribution in each group (suture versus coupler) from the included studies. The majority of the coupler cases come from one study. The authors do investigate risk of bias of their included studies, but highlighting this in their discussion/limitations would be beneficial.

2. Amongst the included studies, can the authors comment on whether these studies noted any patients that were excluded from use of coupler device and reasons (e.g. vessel size mismatch, etc.)?

3. There are minor typos and grammatical errors in the manuscript. I would recommend careful proofreading.

Reviewer #2: This manuscript presents a meta-analysis of retrospective series describing free-flap outcomes with an arterial coupler technique versus and traditional hand-sewn technique. The authors perform a comprehensive review of retrospective studies that describe free-flap outcomes using both techniques. They report results on time to perform anastomosis and the incidence of microvascular thrombosis / vascular compromise. The authors find that the arterial coupler technique, while offering significantly reduced operative time may be associated with a higher rate of thrombosis, though this second association was not statistically significant.

I find the authors meta-analysis technique to be comprehensive and worth reporting. However, I have two recommendations that could improve the manuscript:

(1) The conclusion that “the application of microvascular anastomotic devices will help decrease the anastomotic time and achieve satisfactory vessel patency” is too strong and should either be tempered or removed. While the results do suggested shorter time associated with use of the device, the strong trend toward increased thrombosis is concerning and needs to be further studied. It is certainly not clear that these techniques will be embraced as hand-sewn anastomosis still remains the widely accepted standard.

(2) A comparison of patients who underwent anastomosis with a coupler versus hand-sewn analysis would be an important in evaluating whether these were well matched groups. I recommend a table of patient and tumor characteristics broken down by coupler versus hand-sewn technique and a statistical comparison of each of these categories. Given the retrospective nature of these studies, it would be important to evaluate whether there were any biases in cases with microvascular coupler, i.e more salvage, higher stage, etc….

6. PLOS authors have the option to publish the peer review history of their article (what does this mean?). If published, this will include your full peer review and any attached files.

Reviewer #1: No

Reviewer #2: No

---

## [Author Response · Author response to Decision Letter 0]

28 Feb 2021

Dear the Academic Editor Peter Dziegielewski and Reviewers, 

Thank you for your letter and for the reviewers’ comments concerning our manuscript entitled “Meta-analysis of arterial anastomosis techniques in head and neck free tissue transfer” (PONE-D-20-38890). Those comments are all valuable and very helpful for revising and improving our paper, as well as the important guiding significance to our manuscript. We have studied comments carefully and have made corrections which we hope to meet with the final approval. We marked all changes with the use of “tracked changes”. The main corrections in the paper and point-by-point responses to the reviewers’ comments are listed as following:

Academic Editor asked the following question: 

1. Was there any data on which arteries were used as donor arteries in the neck? Are some better than others?

Response: According to the data from included studies, the facial artery was the most common recipient artery, followed by the superior thyroid artery. But there was no comparison data.

2. Was there any data compare types of flaps used?

 Response: No. All the included studies described types of flaps in their clinical series, but there was no comparison data.

3. Was there any data on when the arterial thrombosis occurred? Immediate? < 24 hrs? > 24 hrs?

 Response: There are three studies (Chernichenko N., Yang Z.Y., Guo Z.) reporting the data on when the arterial thrombosis occurred with a total of 5 cases. Three cases of arterial thrombosis occurred within 24 hours postoperatively, and two cases occurred after 2 days postoperatively.

4. Was there any data on the size of coupler used in failed anastomoses?

 Response: There are two studies (Chernichenko N., Yang Z.Y.) reporting the data on the size of coupler used in failed anastomoses with a total of 4 cases: 2 cases in 2.5mm, 1 in 1.5mm, 1 in 3mm.

The Editor suggested that “Please revise the manuscript to rephrase the duplicated text, cite your sources, and provide details as to how the current manuscript advances on previous work.”

Response: We feel so sorry about text overlap. A language editor have rephased the duplicated text and made significant modifications on illustrations throughout the manuscript. 

Reviewer #1:

Thanks very much for your appreciation and affirmation about our work. We have made appropriate corrections to improve our manuscript. We addressed all the points as following.

1. The reviewer pointed out that “My primary concern is the imbalance of patient distribution in each group (suture versus coupler) from the included studies. The majority of the coupler cases come from one study. The authors do investigate risk of bias of their included studies, but highlighting this in their discussion/limitations would be beneficial”. 

 Response: Thanks very much for your good suggestion. We highlighted this point in the Limitations.

2. The reviewer pointed out that “Amongst the included studies, can the authors comment on whether these studies noted any patients that were excluded from use of coupler device and reasons (e.g. vessel size mismatch, etc.)”.

 Response: We illustrated the contraindication of the use of arterial coupler in Discussion section. According to the literatures, the contraindications for application of microvascular anastomotic devices in arterial anastomosis are concluded as following: Firstly, the caliber of donor and recipient arteries are less than 1 mm. Secondly, the discrepancies in the arterial luminal diameter of the donor and recipient vessels are greater than a 1.5:1 ratio. Lastly, there is severe fibrosis and atherosclerotic plaque within the artery. 

3. The reviewer commented that “There are minor typos and grammatical errors in the manuscript. I would recommend careful proofreading”.

Response: We feel so sorry about the typos and grammatical errors in the manuscript. We have proofread the full text and corrected those.

Reviewer #2:

Thank you for your review. We have taken into full consideration your advice and made plenty of appropriate corrections. We addressed all the points as following.

1. The reviewer pointed out that “The conclusion that ‘the application of microvascular anastomotic devices will help decrease the anastomotic time and achieve satisfactory vessel patency’ is too strong and should either be tempered or removed. While the results do suggested shorter time associated with use of the device, the strong trend toward increased thrombosis is concerning and needs to be further studied. It is certainly not clear that these techniques will be embraced as hand-sewn anastomosis still remains the widely accepted standard”.

Response: Thank you for your good comment. We have modified the conclusion to make it more legible and accurate.

2. The reviewer suggested that “A comparison of patients who underwent anastomosis with a coupler versus hand-sewn analysis would be an important in evaluating whether these were well matched groups. I recommend a table of patient and tumor characteristics broken down by coupler versus hand-sewn technique and a statistical comparison of each of these categories. Given the retrospective nature of these studies, it would be important to evaluate whether there were any biases in cases with microvascular coupler, i.e more salvage, higher stage, etc….”.

 Response: Thanks for your good suggestion. Because all the cases of arterial thrombosis or arterial compromise were in the microvascular anastomotic group, a table could not be listed to compare the demographics of patients broken down by coupler versus hand-sewn technique. Only one study (Chernichenko N.) gave detailed presentations of flap failures. Imbalanced distribution of patients in the microvascular anastomotic device group and hand-sewn technique group from the included studies might indeed result in bias of this meta-analysis. We highlighted this point in the Limitations.

In summary, we have revised the paper according to the editor and reviewers. Thanks for all the help. We are looking forward to your final acceptance.

---

## [Editor Report · Decision Letter 1]

18 Mar 2021

Meta-analysis of arterial anastomosis techniques in head and neck free tissue transfer

PONE-D-20-38890R1

Dear Dr. Gao,

We’re pleased to inform you that your manuscript has been judged scientifically suitable for publication and will be formally accepted for publication once it meets all outstanding technical requirements.

Kind regards,

Peter Dziegielewski, MD, FRCSC

Academic Editor

PLOS ONE

Additional Editor Comments (optional):

Thank you for your revisions. Well done
---

## [Editor Report · Acceptance letter]

22 Mar 2021

PONE-D-20-38890R1 

Meta-analysis of arterial anastomosis techniques in head and neck free tissue transfer 

Dear Dr. Gao:

I'm pleased to inform you that your manuscript has been deemed suitable for publication in PLOS ONE. Congratulations! Your manuscript is now with our production department. 

Kind regards, 

on behalf of

Dr. Peter Dziegielewski 

Academic Editor

PLOS ONE